# Impact of malaria diagnostic choice on monitoring of *Plasmodium falciparum* prevalence estimates in the Democratic Republic of the Congo and relevance to control programs in high-burden countries

Alpha Oumar Diallo[1], Kristin Banek[2], Melchior Mwandagalirwa Kashamuka[3], Joseph Alexandre Mavungu Bala[3], Marthe Nkalani[3], Georges Kihuma[3], Tommy Mambulu Nseka[4], Joseph Losoma Atibu[3], Georges Emo Mahilu[3], Lauren McCormick[1], Samuel J. White[2], Rachel Sendor[1], Cyrus Sinai[5], Corinna Keeler[1], Camelia Herman[6], Michael Emch[1,5], Eric Sompwe[4], Kyaw Lay Thwai[1], Rhoel R. Dinglasan[7], Eric Rogier[6], Jonathan J. Juliano[1,2], Antoinette Kitoto Tshefu[3‡], Jonathan B. Parr[2‡]*

1 Department of Epidemiology, Gillings School of Global Public Health, University of North Carolina at Chapel Hill, Chapel Hill, North Carolina, United States of America, 2 Institute for Global Health and Infectious Diseases, University of North Carolina at Chapel Hill, Chapel Hill, North Carolina, United States of America, 3 Ecole de Santé Publique, Faculté de Médecine, University of Kinshasa, Kinshasa, Democratic Republic of the Congo, 4 Programme National de Lutte Contre Paludisme, Kinshasa, Democratic Republic of the Congo, 5 Department of Geography, University of North Carolina at Chapel Hill, Chapel Hill, North Carolina, United States of America, 6 Malaria Branch, Division of Parasitic Diseases and Malaria, Center for Disease Control and Prevention, Atlanta, Georgia, United States of America, 7 University of Florida Emerging Pathogens Institute, Department of Infectious Diseases & Immunology, College of Veterinary Medicine, Gainesville, Florida, United States of America

‡ AKT and JBP are co-senior authors on this work.
* jonathan_parr@med.unc.edu

## Abstract

Malaria programs rely upon a variety of diagnostic assays, including rapid diagnostic tests (RDTs), microscopy, polymerase chain reaction (PCR), and bead-based immunoassays (BBA), to monitor malaria prevalence and support control and elimination efforts. Data comparing these assays are limited, especially from high-burden countries like the Democratic Republic of the Congo (DRC). Using cross-sectional and routine data, we compared diagnostic performance and *Plasmodium falciparum* prevalence estimates across health areas of varying transmission intensity to illustrate the relevance of assay performance to malaria control programs. Data and samples were collected between March–June 2018 during a cross-sectional household survey across three health areas with low, moderate, and high transmission intensities within Kinshasa Province, DRC. Samples from 1,431 participants were evaluated using RDT, microscopy, PCR, and BBA. *P. falciparum* parasite prevalence varied between diagnostic methods across all health areas, with the highest prevalence estimates observed in Bu (57.4–72.4% across assays), followed by Kimpoko (32.6–53.2%), and Voix du Peuple (3.1–8.4%). Using latent class analysis to compare these diagnostic methods against an "alloyed gold standard," the most sensitive diagnostic method was BBA in Bu (high prevalence) and Voix du Peuple (low prevalence), while PCR diagnosis was most sensitive in Kimpoko

**Data Availability Statement:** De-identified data analyzed as part of this study is available through the Carolina Digital Repository at the University of North Carolina at Chapel Hill: https://cdr.lib.unc.edu/concern/data_sets/0c483v220?locale=en.

**Funding:** This work was funded by NIH R01AI132547 to JJJ and RRD and R01AI129812 to AT. It was also partially supported by NIH K24AI134990 to JJJ; UNC Training in Infectious Disease Epidemiology (TIDE) T32 training grant from the NIH through the NIAID (T32AI070114) to KB and RS; NIH Research Training Grant (D43009340), funded by the NIH Fogarty International Center, NHBLI, NINDS, NCI, NINR, NIAID, and NIEHS, to KB; and from the American Society of Tropical Medicine and Hygiene/ Burroughs Wellcome Fund to JBP. The funders had no role in study design, data collection and analysis, decision to publish, or preparation of the manuscript.

**Competing interests:** I have read the journal's policy and the authors of this manuscript have the following competing interests: JBP reports financial support from Gilead Sciences, non-financial support from Abbott Diagnostics, and consulting for Zymeron Corporation, all outside the scope of this work. All other authors have declared that no competing interests exist.

(moderate prevalence). RDTs were consistently the most specific diagnostic method in all health areas. Among 9.0 million people residing in Kinshasa Province in 2018, the estimated *P. falciparum* prevalence by microscopy, PCR, and BBA were nearly double that of RDT. Comparison of malaria RDT, microscopy, PCR, and BBA results confirmed differences in sensitivity and specificity that varied by endemicity, with PCR and BBA performing best for detecting any *P. falciparum* infection. Prevalence estimates varied widely depending on assay type for parasite detection. Inherent differences in assay performance should be carefully considered when using community survey and surveillance data to guide policy decisions.

## Introduction

Malaria programs rely upon various diagnostic assays to monitor malaria prevalence and to support control and elimination efforts. Though malaria rapid diagnostic tests (RDTs) and microscopy historically account for nearly all point-of-care clinical *Plasmodium falciparum* diagnoses and surveillance methods in endemic regions, high-throughput polymerase chain reaction (PCR) assays and bead-based immunoassays (BBA) are gaining traction in large surveys worldwide.

Unique features of each assay and the targets they detect are known to influence their performance characteristics in different settings. For example, microscopy or PCR detection is largely limited to the period of active infection, whereas RDT or immunoassay detection of *P. falciparum* histidine-rich protein 2 (HRP2) can extend weeks after clearance of infection because antigen often lingers in the blood [1]. These differences can be beneficial or serve as limitations depending on the question at hand and the epidemiological setting. Prior comparisons of RDT, microscopy, and PCR suggest that local malaria transmission intensity is an important determinant of their diagnostic performance [2]. However, data comparing all four assays–RDT, microscopy, PCR, and BBA is limited, especially from high-burden countries.

The World Health Organization recently called for renewed efforts to address malaria in high-burden countries. 70% of the world's malaria cases are concentrated in only 11 countries, [3] 10 of which are in sub-Saharan Africa. All of these countries reported increases in malaria cases in 2020, continuing a trend that started well before the COVID-19 pandemic [4]. Efforts to move away from "one size fits all" approaches and toward interventions tailored to the local epidemiology are required to achieve sustained progress against malaria [3].

This is especially true in the Democratic Republic of the Congo (DRC), where approximately 12% of global malaria cases and 26 million annual infections occur across a diverse landscape of malaria transmission intensities [4]. Leveraging samples collected during the baseline survey of a longitudinal malaria transmission study in Kinshasa Province [5, 6], we sought to compare malaria diagnostic performance for surveillance in sites with low, moderate, and high *P. falciparum* parasite prevalence and across all ages. We observed differences in performance by test type and modeled the impact of diagnostic assay choice on monitoring province-level prevalence estimates, emphasizing the importance of local context when choosing or interpreting results of malaria diagnostic assays.

## Methods

### Ethics statement

Written formal consent was obtained for all participants except for children, for whom written assent and formal parental consent was obtained. This work was approved by the Kinshasa

School of Public Health Ethics Committee (ESP/CE/021/2017) and the University of North Carolina at Chapel Hill Institutional Review Board (17–1588).

## Data source and study population

We performed all four malaria diagnostic assays–RDT, microscopy, PCR, and BBA–on blood samples and data collected in March-June 2018 (rainy season) as part of a prospective, longitudinal study of malaria conducted in urban and rural communities in Kinshasa Province. The first phase of the cohort was conducted from 2015–2017 [5, 6] followed by a second phase conducted from 2018–2022. This cross-sectional sub-study was conducted among samples from individuals during the second phase baseline household visit.

Detailed participant sampling and enrollment methods have been described previously [5, 6]. Briefly, two health zones within Kinshasa Province were selected based on historic malaria prevalence, ecological diversity to cover the urban/rural gradient, and accessibility year-round by the research team. Bu and Kimpoko health areas, nested within Maluku 1 health zone (rural), were selected to represent high and moderate malaria endemicity settings, respectively (**Fig 1**). The Voix du Peuple health area, within Lingwala health zone (urban), serves as a low malaria endemicity setting and is located within the Kinshasa metropolitan region, the DRC's capital and one of the world's largest and fastest-growing cities [7].

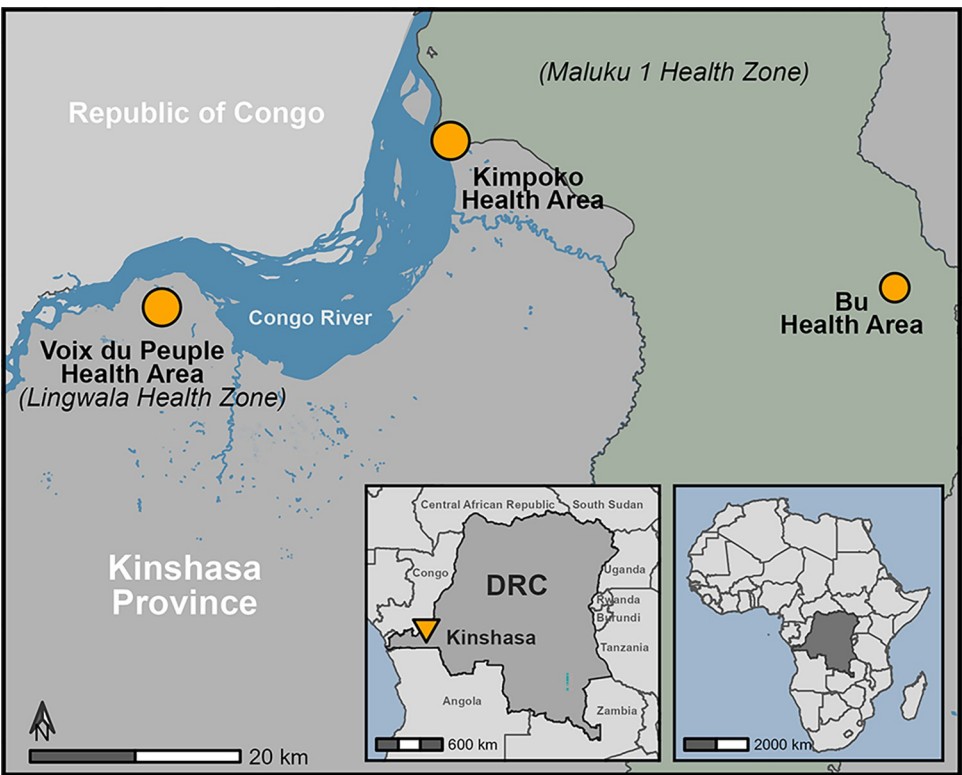

**Fig 1. Location of study sites by health area.** Three rural villages in Bu health area, three peri-urban villages in Kimpoko health area, and one urban neighborhood in Voix du Peuple health area were included, all located within Kinshasa Province. The base map shapefiles for Africa and the DRC were publicly sourced from OpenStreetMap, accessed via the Humanitarian Data Exchange (data.humdata.org): Africa—https://data.humdata.org/dataset/global-lsib-polygons-detailed; DRC rivers (hotosm_cod_waterways_polygons.shp)—https://data.humdata.org/dataset/hotosm_cod_waterways; DRC Health Zones (file: OSM_RDC_sante_zones_211212.shp)—https://data.humdata.org/dataset/zones-de-sante-rdc.

All persons aged six months or older that normally lived in the household were eligible for inclusion. Eligible individuals provided informed consent or minor assent (individuals aged 7–17 years) administered in Lingala or French before enrollment in the second phase. We restricted this analysis to participants with results from all four malaria diagnostic assays– RDT, microscopy, PCR, and BBA. Every participant provided informed consent for study participation and for collection, storage, and use of biological specimens. In addition to parental consent, all children 7–17 years old provided written assent to participate. The study was approved by the Ethical Committee of the Kinshasa School of Public Health (# ESP/CE/021/ 2017) and the University of North Carolina (UNC) Institutional Review Board (# 17–1588) and determined to be an activity not involving human subject research by the Human Subjects office of the Centers for Disease Control and Prevention (project 0900f3eb81bec92c).

## Data collection

At the baseline visit of the second phase study, trained staff administered a questionnaire derived from the Demographic and Health Surveys (MEASURE-DHS, Rockville, MD). Household heads were asked questions on household characteristics to assess household wealth (i.e., housing materials, asset ownership, electrification, and water source). In addition, respondents were asked to provide individual demographic information, bednet ownership, age, and usage, and basic health information related to malaria, such as having a prior malaria diagnosis, medications received, and presence of a fever.

## Rapid diagnostic test (RDT) and microscopy diagnosis

Following completion of the questionnaire, a laboratory technician collected whole blood by finger- or heel-prick from household participants for RDT, slide preparation, and dried blood spot (DBS) collection. During the household visit, whole blood was applied to an SD Bioline Malaria Ag P.f./Pan-pLDH RDT (05FK60, Alere, Gyeonggi-do, Republic of Korea), and results were interpreted on-site according to the manufacturer's protocol. This RDT detects *P. falciparum* histidine-rich protein 2 (HRP2) and pan-Plasmodium lactate dehydrogenase (pLDH) antigens. If positive by either RDT band, the participants were referred to their local health center and treated according to national guidelines. Next, a laboratory technician prepared a thick smear slide labeled with participant codes and the date and time of collection. In brief, a small drop of finger- or heel-prick blood was spread on glass slide and allowed to air dry and stained with 3% Giemsa (Merck, Rahway, NJ, USA; or VWR, Radnor, PA, USA) for 30 minutes. Slides were transported to the study laboratory in Kinshasa, read once by expert microscopists, with review by a second microscopist any time the initial microscopist observed a form of parasite not clear or familiar to them. All slides were read until at least 200 white blood cells (WBCs) were counted. For slides with <100 asexual parasites observed after counting 200 WBCs, review continued until 500 total WBCs had counted. Microscopy parasite densities were estimated using the equation: (number of counted parasites x 8,000)/(number of counted WBCs).

## Sample processing and *P. falciparum* polymerase chain reaction (PCR)

Dried blood spot samples (DBS) were collected onto Whatman filter paper (GE Healthcare, Chicago, IL, USA), dried at ambient temperature, and stored with desiccant at -20˚C in Kinshasa until shipment to UNC for processing and PCR testing [8]. DNA used for PCR was extracted from three 6mm punches per subject using Chelex 100 resin and Tween, as previously described [9]. The real-time PCR assay targets the P. *falciparum*-specific lactate dehydrogenase (*pfldh*) gene with a lower limit of detection of 5–10 parasites/μl [10, 11]. The

quantitative PCR assay was performed using reaction conditions, primers, and quality control measures for high-throughput PCR as previously described, with the exception that assays were performed in singleton and amplification by 40 cycles was considered positive [5]. Primer sequences and reaction conditions are provided in **S1 Table**.

## Multiplex bead-based immunoassay

A single 6mm DBS punch was subjected to a multiplex BBA for detecting *Plasmodium* antigens at the US Centers for Disease Control and Prevention (CDC) as described previously [12] and rehydrated in a buffer with PBS (pH 7.2), 0.3% Tween-20, 0.5% casein, 0.5% BSA, 0.5% polyvinyl alcohol, 0.8% polyvinlypyrrolidine, 0.02% sodium azide, and 3 μg/mL of *E. coli* lysate (to prevent nonspecific binding). Though only HRP2 antigen positivity is utilized for analysis here, the standard BBA multiplex panel was utilized and performed in singleton for this sample set as described below. Magnetic microbeads (Luminex Corp., Austin, TX, USA) were conjugated to antigen capture antibodies by an antibody coupling kit (Luminex Corp.) according to manufacturer's instructions. For one milliliter of microbeads ($12.5 \times 10^6$ beads) antibody coupling concentrations were: anti-pan-*Plasmodium* aldolase antibody (pAldolase, 12.5 μg, Abcam); anti-pan-*Plasmodium* lactate dehydrogenase antibody (pLDH, 12.5 μg of clone M1209063, Fitzgerald); anti-*P. vivax* LDH antibody (PvLDH, 12.5 μg of clone M1709Pv2); anti-*P. falciparum* histidine-rich protein 2 (HRP2, 20 μg, clone MPFG-55A, ICL Inc, Portland, OR, USA). Detection antibodies were also prepared in advance by biotinylating (EZ-link Micro Sulfo-NHS-Biotinylation Kit, Thermo Fisher Scientific, Waltham, MA) according to manufacturer's instructions. Final prepared dilution of detection antibodies was 1.0 mg/mL and for anti-malarial antigen specific antibodies as follows: pAldolase (Abcam, Cambridge, UK), pLDH and PvLDH (1:1 antibody mixture of clones M1709Pv1 and M86550, Fitzgerald Industries, Acton, MA, USA), HRP2 (1:1 antibody mixture of MPFG-55A and MPFM-55A, ICL Inc). Upon conjugation or biotinylation, reagents were stored at 4˚C until use in the immunoassay.

BBA reagents were diluted in buffer: containing 0.1 M phosphate buffered saline (PBS) pH 7.2, 0.05% Tween-20, 0.5% bovine serum albumin (BSA), and 0.02% sodium azide. For all wash steps, assay plate was affixed to a handheld magnet (Luminex Corp), and gently tapped for 2 min to allow bead magnetization before evacuation of liquid and washing with 150 μL PBS, 0.05% Tween-20. The four bead regions were combined in dilution buffer (in a reagent trough) and pipetted onto a 96-well assay plate (BioPlex Pro, Bio-Rad, Hercules, CA, USA) at a quantity of approximately 800 beads/region. Plates were washed twice, and 50 μL of controls or samples pipetted into appropriate wells. Following 90-min gentle shaking at room temperature protected from light, plates were washed three times. A mixture of detection antibodies was prepared in dilution buffer (pAldolase at 1:2000, all others at 1:500), and 50 μL added to each well for a 45-min incubation. After three washes, 50 μL of streptavidin–phycoerythrin (at 1:200, Invitrogen) was added for a 30-min incubation. Plates washed three times, and 50 μL dilution buffer added to each well for 30-min incubation. Plates washed once and beads resuspended in 100 mL PBS. After brief shaking, plates were read on MAGPIX machine (R&D Systems, Minneapolis, MN, USA) with target of 50 beads per region. The median fluorescence intensity (MFI) value was generated for all beads collected for each region by assay well and subtracting the assay signal from wells with dilution buffer blank provides an MFI-background (MFI-bg) value used for analyses. Positive and negative controls were included on each plate to ensure assay performance.

## Statistical analyses

We estimated *P. falciparum* malaria prevalence and 95% confidence intervals (CIs) across health areas, age categories, sex, self-reported history of fever in the last seven days, and household

membership size from RDT, microscopy, PCR, and BBA results based on predicted values using univariate Generalized Estimating Equations (GEE) logistic regression. The GEE yields population-average estimates while accounting for the potential influence of sharing a household with a malaria-infected person on other members' infection status [13]. To model the correlation between household members, we assumed an exchangeable working correlation structure [14].

We estimated sensitivity and specificity using the classical contingency table approach and latent class analysis (LCA) to evaluate the performance of the four diagnostic methods (i.e., RDT, microscopy, PCR, and BBA) to detect *P. falciparum* malaria infection. In the contingency table approach, we used PCR or BBA as the reference ("gold standard") because they are highly sensitive assays that detect different targets (nucleic acid and antigen) and outperform the traditional microscopy gold standard in most settings. Given that there is not a clear gold standard, we also used LCA to estimate sensitivity and specificity by combining results from the four diagnostic methods via a probabilistic model to define an internal reference standard, or "alloyed gold standard" [12, 15–17].

To inform the potential impact of malaria diagnostic method choice on malaria control programs–for example, during decision-making about RDT or antimalarial medication procurement–we estimated the number of individuals with *P. falciparum* malaria infection in Kinshasa Province during the phase two baseline data collection period (March-June 2018) using each diagnostic method: RDT, microscopy, *pfldh* real-time PCR, and BBA. First, we stratified health areas in Kinshasa Province into three malaria prevalence categories: prevalence of 15–24% (low), 25–34% (moderate), or ≥35% (high) using PCR-based prevalence estimates from the 2013–2014 Demographic and Health Survey (DHS) [18]. We modelled Kinshasa Province malaria prevalence by calculating the cluster-level malaria prevalence based on all samples for each DHS cluster. Then, we fit a thin plate spline model to the cluster-level prevalence values to spatially interpolate the malaria prevalence across Kinshasa Province in areas without DHS clusters [19]. Finally, we used zonal statistics to estimate the mean malaria prevalence in each health area within Kinshasa Province. While there are 416 health areas in Kinshasa Province, we could only estimate the mean malaria prevalence for 402 (96.6%) health areas, where health area population estimates from the Ministry of Health were able to be linked by name to corresponding health area boundaries from a publicly available geospatial dataset [20].

Second, each health area in Kinshasa Province was matched with the study health area with the most similar *P. falciparum* prevalence. Third, we estimated age-specific malaria prevalence estimates by RDT, microscopy, PCR, and BBA from the study health areas. For this analysis, we stratified age into two categories, <5 and ≥5 years old, using age data provided by the DRC Ministry of Health. Fourth, we generated age-specific weights by multiplying the study age-specific prevalence estimates and age distribution (<5 years: 18.9%; ≥5 years: 81.1%) used by the DRC Ministry of Health. Fifth, we multiplied the age-specific weights by the age-stratified population estimates provided by the DRC Ministry of Health for 2018 and obtained from the DRC District Health Information System 2 (DHIS2) to calculate the age-specific number of individuals with malaria by diagnostic method for 402 health areas in Kinshasa Province [21]. Finally, we summed the estimated number of cases by diagnostic method to obtain estimates for the entire province. See **S1 Text** for a detailed description of these methods. We conducted analyses in R Statistical Software (v4.1.2; [22]).

## Results

### Study participant description

Among the 1,450 individuals across 226 households with samples collected in March-June 2018 who underwent malaria laboratory testing, we excluded 19 (1.3%) individuals for missing

**Table 1. Malaria prevalence by diagnostic method (RDT, microscopy, PCR, and BBA) for the detection of *Plasmodium falciparum* (N = 1,431).**

| Characteristics | Study population N (%)[a] | *P. falciparum* prevalence (95% CI)[b] | | | |
|---|---|---|---|---|---|
| | | RDT[c] | Microscopy[d] | PCR[e] | BBA[f] |
| Overall | 1,431 | 33.7 (29.9–37.8) | 43.3 (39.3–47.5) | 46.1 (41.8–50.6) | 48.4 (44.0–52.8) |
| Health area | | | | | |
| Bu<br>All age groups | 491 (34.3) | 57.4 (51.7–62.9) | 59.7 (54.0–65.1) | 66.2 (60.2–71.8) | 72.4 (66.9–77.2) |
| <5 | 99 (20.2) | 60.6 (50.3–70.0) | 53.4 (43.2–63.4) | 50.3 (39.9–60.6) | 72.6 (62.4–80.9) |
| 5–14 | 196 (39.9) | 77.2 (69.9–83.1) | 71.4 (63.0–78.6) | 77.6 (69.5–84.0) | 85.8 (78.1–91.1) |
| 15–24 | 47 (9.6) | 43.6 (29.9–58.3) | 68.3 (54.3–79.6) | 76.7 (64.2–85.7) | 69.2 (54.0–81.1) |
| ≥ 25 | 149 (30.3) | 34.9 (27.3–43.4) | 46.0 (37.9–54.4) | 59.0 (50.3–67.2) | 56.3 (47.7–64.4) |
| Kimpoko<br>All age groups | 521 (36.4) | 32.6 (28.2–37.4) | 50.7 (45.5–55.9) | 53.2 (47.8–58.5) | 51.2 (45.8–56.6) |
| <5 | 61 (11.8) | 16.3 (8.8–28.2) | 38.5 (27.7–50.5) | 29.6 (19.8–41.8) | 32.3 (22.2–44.4) |
| 5–14 | 190 (36.6) | 52.5 (43.9–60.9) | 60.6 (52.3–68.4) | 66.4 (57.9–73.9) | 61.4 (52.7–69.5) |
| 15–24 | 98 (18.9) | 38.6 (29.7–48.3) | 59.7 (49.5–69.1) | 68.1 (58.6–76.3) | 59.4 (49.6–68.4) |
| ≥ 25 | 170 (32.8) | 14.0 (9.6–20.0) | 39.8 (32.4–47.7) | 39.2 (31.6–47.3) | 42.0 (34.7–49.7) |
| Voix du Peuple<br>All age groups | 419 (29.3) | 3.1 (1.7–5.5) | 7.6 (5.1–11.2) | 5.6 (3.5–8.8) | 8.4 (6.0–11.6) |
| <5 | 36 (8.6) | 0.0 (0.0–0.0) | 16.3 (6.3–36.0) | 2.1 (0.3–14.5) | 0.0 (0.0–0.0) |
| 5–14 | 101 (24.1) | 5.9 (2.5–13.5) | 7.6 (3.4–16.1) | 8.7 (4.2–17.2) | 9.8 (5.3–17.5) |
| 15–24 | 119 (28.4) | 3.4 (1.3–8.1) | 9.1 (5.1–15.7) | 8.1 (4.6–14.1) | 10.9 (6.7–17.3) |
| ≥ 25 | 163 (38.9) | 2.5 (1.1–5.2) | 5.5 (3.2–9.3) | 3.0 (1.3–6.9) | 7.4 (4.2–12.9) |
| Sex[g] | | | | | |
| Male | 648 (45.3) | 34.7 (30.4–39.4) | 44.7 (40.0–49.5) | 48.4 (43.2–53.7) | 48.3 (43.1–53.6) |
| Female | 783 (54.7) | 32.9 (28.6–37.4) | 42.2 (37.5–47.0) | 44.2 (39.6–48.9) | 48.5 (43.7–53.3) |
| Self–reported fever in last seven days | | | | | |
| No | 1,137 (83.7) | 31.2 (27.3–35.4) | 41.8 (37.5–46.2) | 45.6 (41.0–50.2) | 47.0 (42.3–51.7) |
| Yes | 222 (16.3) | 44.6 (37.3–52.1) | 50.2 (42.2–58.2) | 49.2 (41.4–57.0) | 55.0 (47.4–62.4) |
| Household members enrolled | | | | | |
| 1–5 | 320 (22.4) | 35.8 (29.4–42.8) | 49.5 (42.5–56.6) | 51.4 (44.3–58.6) | 55.0 (47.5–62.2) |
| 6–10 | 844 (59.0) | 35.0 (29.9–40.5) | 41.8 (36.5–47.3) | 46.1 (40.2–52.1) | 47.1 (41.3–53.0) |
| 11–15 | 178 (12.4) | 25.3 (14.8–39.7) | 37.5 (24.7–52.4) | 33.3 (20.1–49.7) | 40.1 (26.1–56.0) |
| 16–20 | 89 (6.2) | 9.1 (2.5–28.5) | 16.8 (6.6–36.8) | 15.7 (5.0–39.6) | 13.5 (3.7–39.0) |

Abbreviations: 95% CI, 95% confidence interval; BBA, bead-based immunoassay; PCR, polymerase chain reaction; RDT, rapid diagnostic test.

[a] Missing: age—Kimpoko (n = 2); self-reported fever in last seven days (n = 72)

[b] Used generalized estimating equation logistic regression to estimate prevalence and 95% CI

[c] HRP2-band positive

[d] Any *Plasmodium* species visualized

[e] *P. falciparum* lactate dehydrogenase PCR-positive

[f] *P. falciparum* HRP2 antigen positive

[g] Self-reported at enrollment

or failed results for at least one of the diagnostic assays—RDT, microscopy, PCR, and BBA. The 1,431 individuals included in this analysis were distributed across the study health areas as follows: 491 (34.3%) in Bu (three rural villages), 521 (36.4%) in Kimpoko (three peri-urban villages), and 419 (29.3%) in Voix du Peuple (one urban neighborhood) (**Table 1**). The median age of subjects was 20.0 years old (interquartile range: 11.0, 39.5), most were female (55.3%), and they shared a household with 6–10 other study participants (59.0%).

### *P. falciparum* infection prevalence

*P. falciparum* parasite prevalence varied between diagnostic methods across all health areas, with the highest prevalence estimates observed in Bu (57.4–72.4% across assays), followed by Kimpoko (32.6–53.2%), and Voix du Peuple (3.1–8.4%), which we defined as "high," "moderate," and "low" prevalence areas, respectively (**Table 1**, **Fig 2A**). *P. falciparum* prevalence was consistently lowest by RDT (HRP2 band) and highest by BBA (HRP2 detection) and PCR (*pfldh* amplification). Across age categories, malaria prevalence was highest among school-aged children (5–14 years) and young adults (15–24 years), participants who self-reported experiencing a fever in the last seven days, and those who lived in households with 1–5 members enrolled.

### Diagnostic performance for the detection of *P. falciparum*

Overall, 888 (62%) samples had perfect concordance (23.3% positive and 38.7% negative) across the four diagnostic assays, with the highest proportion of positive and concordance found Bu (high prevalence) and Voix du Peuple (low prevalence), respectively (**Fig 3**). Diagnostic assay performance for the detection of *P. falciparum* varied by endemicity and gold standard choice (**Table 2** **and S2 Table**). In general, assay sensitivity decreased, and specificity increased for the detection of *P. falciparum* when moving from higher to lower prevalence health areas. Among the three diagnostic methods compared to PCR, BBA (sensitivity 79.5% [95% CI: 76.1–82.6]; specificity 80.2% [95% CI: 77.3–82.9]) had the highest sensitivity but lower specificity than RDT and microscopy across all health areas with the exception of Voix du Peuple (low prevalence). Compared to the BBA, PCR (sensitivity 75.8% [95% CI: 73.2–79.6]; specificity 83.0% [95% CI: 80.2–85.5]) had the highest sensitivity but a lower specificity than RDT.

Using LCA to compare all four diagnostic methods against an "alloyed gold standard," the best diagnostic performance varied by health area (**Fig 2B**). The most sensitive diagnostic method was BBA in Bu (high prevalence) and Voix du Peuple (low prevalence), while PCR was the most sensitive diagnostic method in Kimpoko (moderate prevalence). RDT was consistently the most specific diagnostic method in all health areas, regardless of the gold standard (PCR, BBA, or LCA). We did not observe a single, best-performing assay.

### Estimated *P. falciparum* infection in Kinshasa Province by diagnostic method

*P. falciparum* infections were estimated to be 1,169,745 by BBA, 1,060,485 by PCR, and 1,089,525 by microscopy–all nearly double the 672,431 estimated cases by RDT (**Table 3**). Though imprecise, these numbers approximate how diagnostic choice would influence prevalence estimates across 402 health areas within 35 health zones in Kinshasa Province (**Fig 4**), with an assumed population of 9.0 million in 2018 based on DHIS2 data.

### Discussion

The performance of commonly used malaria diagnostic assays varied based on the epidemiological context in this cross-sectional household study comprising rural, peri-urban, and urban sites with a range of parasite prevalence in a high-burden setting. Compared to an alloyed gold standard, PCR and BBA had higher sensitivity for *P. falciparum* than RDT and microscopy. However, neither assay outperformed the other in all contexts. BBA had the best sensitivity low- and high-prevalence sites, while PCR had the best sensitivity in moderate-

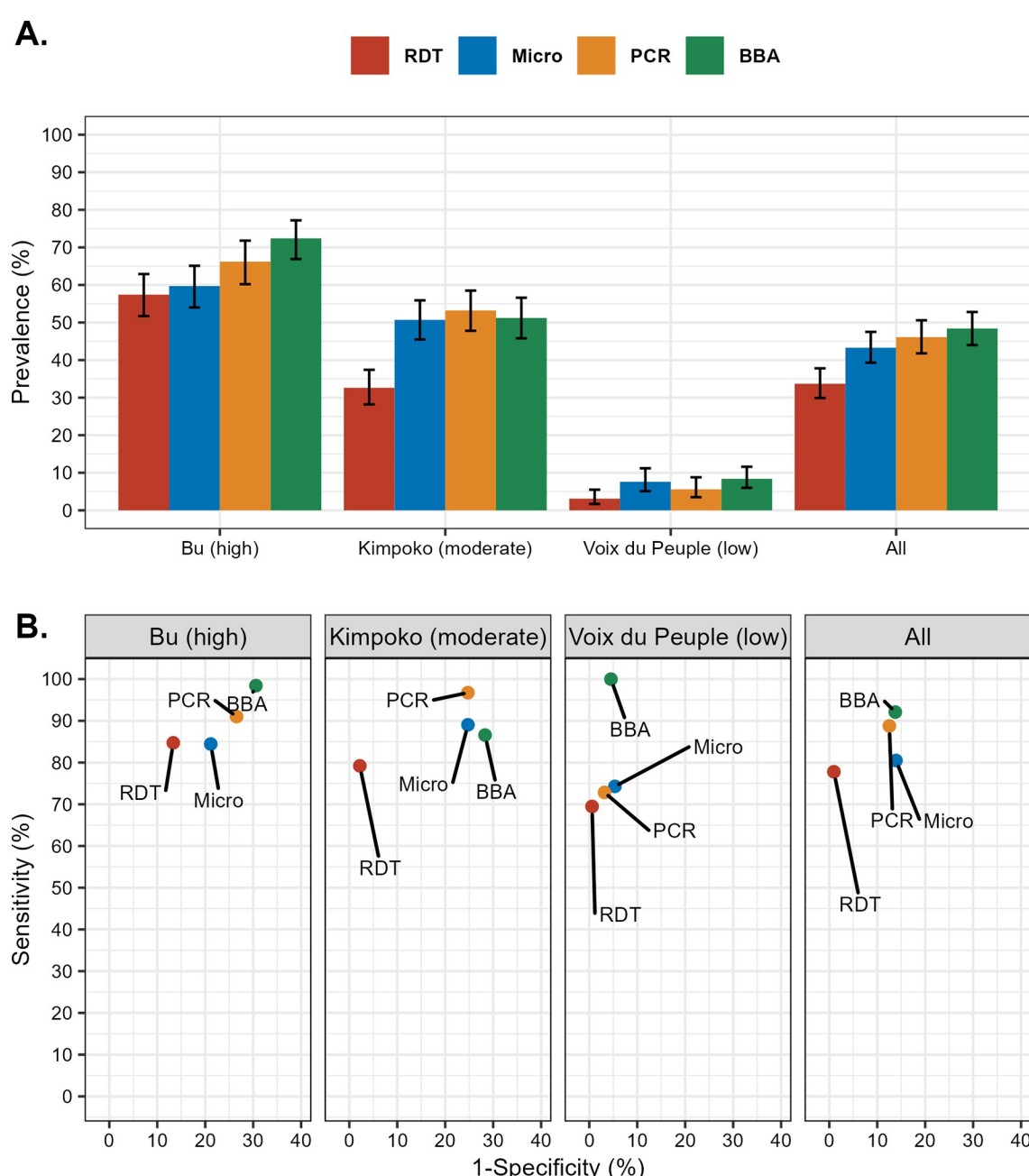

**Fig 2. *P. falciparum* prevalence and diagnostic assay performance.** (A) *P. falciparum* prevalence and 95% confidence intervals across diagnostic assays by health area. (B) Diagnostic assay performance compared to an alloyed gold standard by health area. A diagnostic assay with perfect sensitivity and specificity would fall in the top left corner of panel B. Abbreviations: RDT, rapid diagnostic test HRP2-band; Microscopy, thick-smear light microscopy; PCR, real-time polymerase chain reaction detecting the *pfldh* gene; BBA, bead-based immunoassay detecting HRP2 protein. [a]Error bars are 95% confidence intervals. *Abbreviations*: RDT, rapid diagnostic test HRP2-band; Microscopy, thick-smear light microscopy; PCR, real-time polymerase chain reaction detecting the *pfldh* gene; BBA, bead-based immunoassay detecting HRP2 protein. [a]Error bars are 95% confidence intervals.

prevalence sites. Across all prevalence strata, RDT had the highest specificity but lowest sensitivity of all four assays.

Differences in prevalence estimates were evident across study sites and age strata. Among children younger than 15 years old in Bu health area's high prevalence sites, prevalence

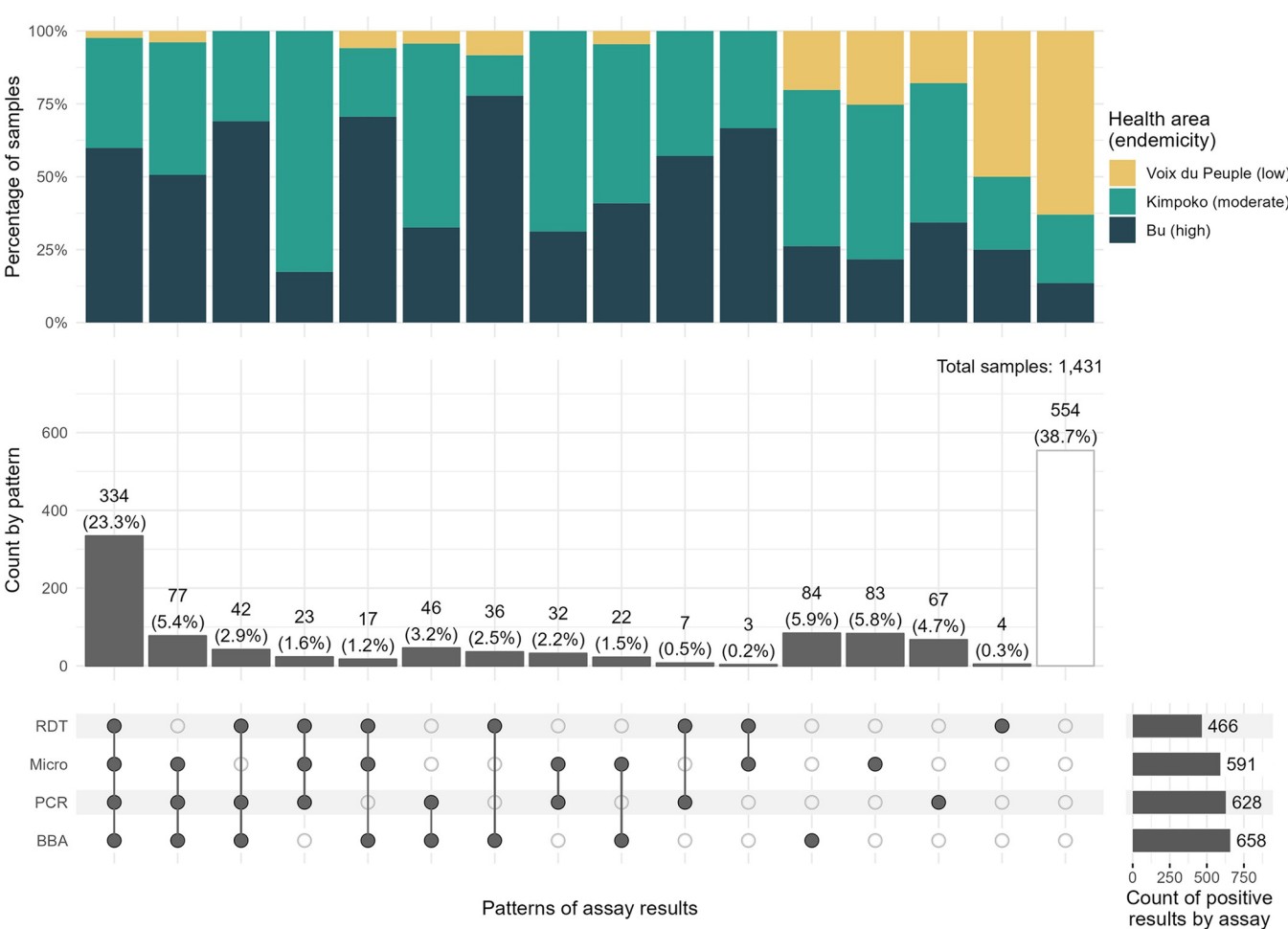

**Fig 3. Patterns of diagnostic assay results and distribution of health area (endemicity) for each pattern.** The top (stacked) bar graph shows the distribution of health area for each pattern intersection or pattern (connecting dots- in bottom figure) across results from the four diagnostic assays. The bottom vertical bar chart displays the count of samples for each pattern and percent of samples with each pattern among the 1,431 samples (vertical bars); the horizontal bars represent the count of positive results by diagnostic assay. *Abbreviations*: RDT, rapid diagnostic test HRP2-band; Microscopy, thick-smear light microscopy; PCR, real-time polymerase chain reaction detecting the *pfldh* gene; BBA, bead-based immunoassay detecting HRP2 protein.

estimates were notably higher by BBA than by PCR. People living in these villages had more frequent infections than in other health areas and, as a result, were expected to be more likely to have lingering HRP2 antigenemia outside the window of acute infection. The highest proportion of recently cleared *P. falciparum* infections (assessed by BBA-positive, *pfldh* PCR-negative) was found in the high-prevalence villages of Bu health area. We also observed a high prevalence of *P. falciparum* infection in 5- to 14-year-olds, consistent with past findings from the DRC [23]. These findings support the argument that older children and adolescents should be routinely included in malaria surveillance activities.

National malaria control programs and their partners rely upon routine data, as well as community-based and health facility surveys, to guide purchasing and forecasting decisions. Parasite prevalence and malaria incidence are usually estimated using the results of RDTs, the primary malaria diagnostic method used for case management in the DRC and across sub-Saharan Africa. However, high-throughput molecular and serological assays are becoming more readily available throughout the region for surveillance [24]. In addition, new "ultrasensitive" rapid diagnostic tests have been developed but are not currently used routinely in

**Table 2. Sensitivity and specificity of malaria diagnostic methods versus PCR, BBA, and LCA for the detection of *Plasmodium falciparum*, stratified by health area and malaria prevalence.**

| Health area *prevalence* | Diagnostic test | Versus PCR | | Versus BBA | | LCA | |
|---|---|---|---|---|---|---|---|
| | | Sensitivity (95% CI) | Specificity (95% CI) | Sensitivity (95% CI) | Specificity (95% CI) | Sensitivity (95% CI) | Specificity (95% CI) |
| Bu *high* | RDT | 73.2 (68.0–77.9) | 74.2 (66.9–80.7) | 76.0 (71.2–80.3) | 92.0 (86.1–95.9) | 84.7 (80.3–89.8) | 86.7 (79.8–93.1) |
| | Micro | 78.1 (73.2–82.5) | 75.4 (68.2–81.8) | 73.4 (68.5–78.0) | 75.2 (67.1–82.2) | 84.4 (79.5–89.4) | 78.8 (71.7–84.7) |
| | PCR | N/A | N/A | 79.9 (75.4–84.0) | 70.1 (61.7–77.6) | 91.0 (87.4–95.1) | 73.5 (65.6–80.9) |
| | BBA | 87.3 (83.2–90.8) | 57.5 (49.6–65.1) | N/A | N/A | 98.4 (95.9–100.0) | 69.4 (62.1–76.8) |
| Kimpoko *moderate* | RDT | 57.7 (51.7–63.6) | 95.4 (92.0–97.7) | 55.0 (48.9–61.1) | 90.5 (86.2–93.8) | 79.2 (71.5–86.0) | 97.8 (95.2–100.0) |
| | Micro | 72.4 (66.8–77.6) | 74.8 (68.8–80.1) | 65.8 (59.8–71.4) | 65.9 (59.7–71.7) | 89.0 (83.2–93.9) | 75.3 (70.0–79.9) |
| | PCR | N/A | N/A | 65.8 (59.8–71.4) | 65.9 (59.7–71.7) | 96.7 (92.9–99.5) | 75.3 (69.4–80.2) |
| | BBA | 72.8 (67.1–77.9) | 72.7 (66.7–78.2) | N/A | N/A | 86.6 (81.1–90.9) | 71.7 (65.5–76.5) |
| Voix du Peuple *low* | RDT | 32.0 (14.9–53.5) | 98.5 (96.7–99.4) | 34.3 (19.1–52.2) | 99.5 (98.1–99.9) | 69.5 (38.2–99.9) | 99.4 (98.7–100.0) |
| | Micro | 44.0 (24.4–65.1) | 94.2 (91.4–96.3) | 37.1 (21.5–55.1) | 94.5 (91.8–96.6) | 74.3 (46.8–100.0) | 94.7 (91.9–97.0) |
| | PCR | N/A | N/A | 37.1 (21.5–55.1) | 96.9 (94.6–98.4) | 72.8 (44.4–100.0) | 96.8 (95.3–98.7) |
| | BBA | 52.0 (31.3–72.2) | 94.4 (91.7–96.5) | N/A | N/A | 100.0 (91.2–100.0) | 95.5 (93.2–98.0) |
| All | RDT | 64.7 (60.8–68.4) | 92.5 (90.5–94.2) | 65.2 (61.4–68.8) | 95.2 (93.5–96.6) | 77.8 (74.5–81.8) | 99.0 (98.2–99.9) |
| | Micro | 74.2 (70.6–77.6) | 84.4 (81.7–86.9) | 68.4 (64.7–71.9) | 81.8 (78.8–84.4) | 80.5 (76.5–84.0) | 86.1 (82.9–88.4) |
| | PCR | N/A | N/A | 75.8 (72.4–79.1) | 83.3 (80.5–85.9) | 88.8 (86.0–91.9) | 87.5 (84.2–90.0) |
| | BBA | 79.5 (76.1–82.6) | 80.2 (77.3–82.9) | N/A | N/A | 92.1 (89.6–94.6) | 86.2 (83.2–88.9) |

*Abbreviations*: 95% CI, 95% confidence interval; BBA, bead-based immunoassay; LCA, latent class analysis; Micro, microscopy; N/A, not applicable; PCR, polymerase chain reaction; RDT, rapid diagnostic test.

Africa. Malaria programs will increasingly have access to results from advanced laboratory approaches and need empirical data to guide their decision-making.

To explore how diagnostic assay choice might impact real programmatic decisions, we modeled health area *P. falciparum* PCR prevalence using results from the national 2013–14 Demographic and Health Survey and extrapolated results from our study sites to estimate prevalence across all of Kinshasa Province. Kinshasa Province is the DRC's most populous administrative region and includes rural, peri-urban, and urban zones, including the capital city Kinshasa. Its diverse health zones thus provide an opportunity to examine how differences in diagnostic assay performance across a variety of epidemiological contexts might impact malaria programs. Among nearly 9.0 million children and adults living across 402 health areas in Kinshasa Province in 2018, based on DHIS2 data, the parasite prevalence estimates ranged from 672,431 (7.5%) to 1,169,745 (13.0%), depending on the diagnostic method.

In practical terms, the numbers used to guide RDT or artemisinin-combination therapy procurement choices could vary by nearly two-fold depending on whether they were derived

**Table 3. Estimated number of individuals infected with *P. falciparum* in Kinshasa Province in March-April 2018 by malaria diagnostic method, derived from Ministry of Health age distribution and population estimates (8,993,453 people).**

| Diagnostic method | Number of infections (95% CI) | Percentage infected (95% CI) |
|---|---|---|
| RDT | 672,431 (515,265–918,597) | 7.5 (5.7–10.2) |
| Microscopy | 1,089,525 (885,581–1,364,353) | 12.1 (9.8–15.2) |
| PCR | 1,060,485 (869,752–1,326,532) | 11.8 (9.7–14.7) |
| BBA | 1,169,745 (1,090,512–1,246,787) | 13.0 (12.1–13.9) |

*Abbreviations*: BBA, bead-based immunoassay; PCR, polymerase chain reaction; RDT, rapid diagnostic test.

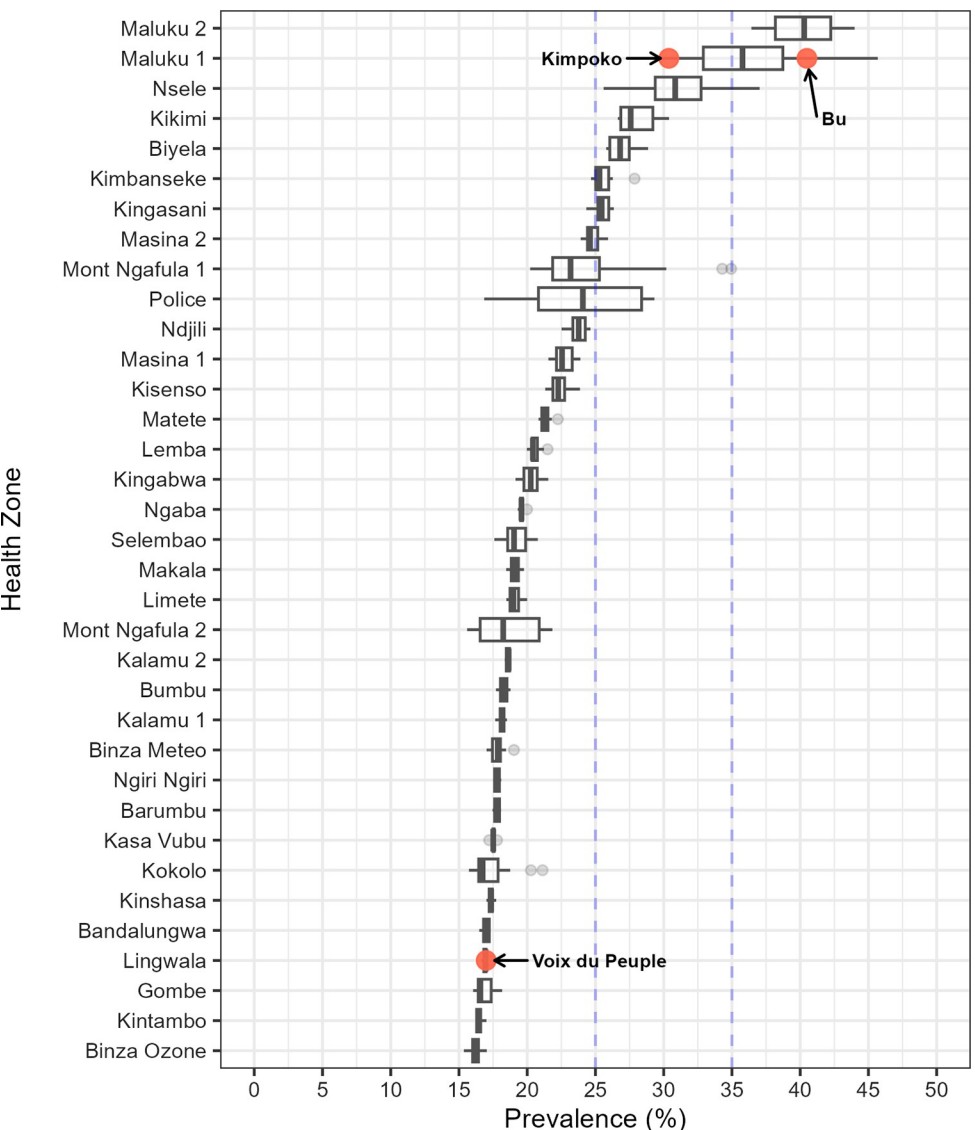

**Fig 4. Distribution of health area malaria prevalence by PCR from the 2013–2014 Demographic Health Survey across health zones in Kinshasa Province.** Representative health areas included in the Kinshasa Province study of four diagnostic assays are indicated by text. The 25% and 35% prevalence thresholds utilized in this study to designate low, moderate, or high prevalence are indicated by vertical hashed lines.

from RDT or BBA results. While these differences may be self-evident to laboratory experts accustomed to interpreting differences between these assays, most programmatic staff are unaccustomed to interpreting these results. Our prevalence estimates are imprecise, but their wide range demonstrates the importance of considering the assay used and its performance characteristics when interpreting malaria survey results across different endemicities.

Our study has several limitations. First, our seven study sites provide valuable insights into malaria epidemiology across different transmission intensities and geographical contexts, but they are not representative of the whole country. The DRC is Africa's second largest country by land mass and the fourth largest by population; it is bordered by nine other countries. For this reason, we restricted our modeling efforts to Kinshasa Province and did not attempt to generalize these findings to the entire country. Second, RDTs were developed for clinical

diagnosis at the point-of-care, and their use for surveillance activities is off-label. This study's household survey design did not allow us to evaluate assay performance during acute malaria infection. People sick with malaria tend to have higher parasite densities that are readily detected by all four assays evaluated. Nonetheless, individual-level data comparing RDT, microscopy, real-time PCR, and BBA results shed light on their relative strengths and weaknesses and can serve as useful data for future modeling efforts. Third, our latent class analysis approach to generate an "alloyed gold standard" assumes conditional independence, but both the RDTs and BBA detect HRP2. This assumption could bias LCA results in favor of both methods [15]. Fourth, survey data utilizing advanced laboratory methodologies may become less important as the DRC National Malaria Control Program, like other malaria programs in Africa, makes increasing use of routine data available through the DHIS2. However, rapid expansion of molecular and serological laboratory capacity across Africa will almost certainly translate to increased use of PCR and BBAs for malaria surveillance during large-scale surveys in the future.

Comparison of RDT, microscopy, real-time PCR, and BBA results confirmed differences in sensitivity and specificity that varied by study site, with PCR and BBA performing best for detecting *P. falciparum* infection. Using a model of the *P. falciparum* prevalence across all of Kinshasa Province and routine data, we found that the number of infections varied nearly two-fold depending on which assay was used for parasite detection. Malaria control programs should carefully consider inherent differences in assay performance when using community survey and surveillance data to guide planning and implementation strategies.

## Supporting information

**S1 Table. Primer sequences and reaction conditions for the real-time PCR assay targeting the *P. falciparum*-specific lactate dehydrogenase (*pfldh*) gene.**
(DOCX)

**S2 Table. Positive and negative predictive values of malaria diagnostic methods versus PCR and BBA for the detection of *Plasmodium falciparum*, stratified by health area and malaria prevalence.**
(DOCX)

**S1 Text. Detailed methods used to estimate the number of *Plasmodium falciparum* infections in Kinshasa Province in March-June 2018 by malaria diagnostic method, derived from the Democratic Republic of the Congo (DRC) Ministry of Health age distribution and population estimates (8,993,453 people).**
(DOCX)

## Acknowledgments

The authors thank Steve Meshnick for conceptualizing the parent study and providing invaluable mentorship relevant to this project before his untimely death. They also thank study participants and the Kinshasa School of Public Health field research team who made this work possible.

## Author Contributions

**Conceptualization:** Alpha Oumar Diallo, Kristin Banek, Melchior Mwandagalirwa Kashamuka, Corinna Keeler, Jonathan B. Parr.

**Data curation:** Alpha Oumar Diallo, Kristin Banek, Melchior Mwandagalirwa Kashamuka, Joseph Alexandre Mavungu Bala, Joseph Losoma Atibu, Georges Emo Mahilu, Lauren McCormick, Samuel J. White, Cyrus Sinai, Kyaw Lay Thwai.

**Formal analysis:** Alpha Oumar Diallo, Corinna Keeler.

**Funding acquisition:** Rhoel R. Dinglasan, Jonathan J. Juliano, Jonathan B. Parr.

**Investigation:** Alpha Oumar Diallo, Joseph Alexandre Mavungu Bala, Marthe Nkalani, Georges Kihuma, Tommy Mambulu Nseka, Joseph Losoma Atibu, Camelia Herman, Kyaw Lay Thwai, Eric Rogier.

**Methodology:** Alpha Oumar Diallo, Kristin Banek.

**Project administration:** Kristin Banek, Joseph Alexandre Mavungu Bala, Marthe Nkalani, Georges Kihuma, Tommy Mambulu Nseka, Joseph Losoma Atibu, Georges Emo Mahilu, Kyaw Lay Thwai, Rhoel R. Dinglasan, Jonathan J. Juliano, Antoinette Kitoto Tshefu, Jonathan B. Parr.

**Resources:** Rhoel R. Dinglasan, Jonathan J. Juliano.

**Supervision:** Antoinette Kitoto Tshefu.

**Validation:** Melchior Mwandagalirwa Kashamuka, Joseph Alexandre Mavungu Bala, Marthe Nkalani, Georges Kihuma, Tommy Mambulu Nseka, Georges Emo Mahilu.

**Visualization:** Alpha Oumar Diallo, Rachel Sendor, Cyrus Sinai.

**Writing – original draft:** Alpha Oumar Diallo, Kristin Banek, Corinna Keeler, Eric Rogier, Jonathan B. Parr.

**Writing – review & editing:** Alpha Oumar Diallo, Kristin Banek, Melchior Mwandagalirwa Kashamuka, Joseph Alexandre Mavungu Bala, Marthe Nkalani, Georges Kihuma, Tommy Mambulu Nseka, Joseph Losoma Atibu, Georges Emo Mahilu, Lauren McCormick, Samuel J. White, Rachel Sendor, Cyrus Sinai, Corinna Keeler, Camelia Herman, Michael Emch, Eric Sompwe, Kyaw Lay Thwai, Rhoel R. Dinglasan, Eric Rogier, Jonathan J. Juliano, Antoinette Kitoto Tshefu, Jonathan B. Parr.

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
