## [Decision Letter · Decision Letter 0]

12 Jan 2023

PGPH-D-22-01821

Impact of malaria diagnostic choice on monitoring of *Plasmodium falciparum* prevalence estimates in the Democratic Republic of the Congo and relevance to control programs in high-burden countries

Dear Dr. Alpha Oumar Diallo,

Thank you for submitting your manuscript to PLOS Global Public Health. After careful consideration, we feel that it has merit but does not fully meet PLOS Global Public Health’s publication criteria as it currently stands. Therefore, we invite you to submit a revised version of the manuscript that addresses the points raised during the review process.

We look forward to receiving your revised manuscript.

Kind regards,

Srinivasa Rao Mutheneni, PhD

Academic Editor

Journal Requirements:

1. Please send a completed 'Competing Interests' statement, including any COIs declared by your co-authors. If you have no competing interests to declare, please state "The authors have declared that no competing interests exist". Otherwise please declare all competing interests beginning with the statement "I have read the journal's policy and the authors of this manuscript have the following competing interests:"

3. In the online submission form, you indicated that your data will be submitted to a repository upon acceptance.  We strongly recommend all authors deposit their data before acceptance, as the process can be lengthy and hold up publication timelines. Please note that, though access restrictions are acceptable now, your entire data will need to be made freely accessible if your manuscript is accepted for publication. This policy applies to all data except where public deposition would breach compliance with the protocol approved by your research ethics board. If you are unable to adhere to our open data policy, please kindly revise your statement to explain your reasoning and we will seek the editor's input on an exemption. Please be assured that, once you have provided your new statement, the assessment of your exemption will not hold up the peer review process.

Please review your reference list to ensure that it is complete and correct. If you have cited papers that have been retracted, please include the rationale for doing so in the manuscript text or remove these references and replace them with relevant current references. Any changes to the reference list should be mentioned in the rebuttal letter that accompanies your revised manuscript. If you need to cite a retracted article, indicate the article’s retracted status in the References list and also include a citation and full reference for the retraction notice.

Additional Editor Comments (if provided):

Reviewers' comments:

Reviewer's Responses to Questions

**Comments to the Author**

1. Does this manuscript meet PLOS Global Public Health’s publication criteria? Is the manuscript technically sound, and do the data support the conclusions? The manuscript must describe methodologically and ethically rigorous research with conclusions that are appropriately drawn based on the data presented.

Reviewer #1: Yes

Reviewer #2: Yes

Reviewer #3: Yes

2. Has the statistical analysis been performed appropriately and rigorously?

Reviewer #1: Yes

Reviewer #2: Yes

Reviewer #3: Yes

3. Have the authors made all data underlying the findings in their manuscript fully available (please refer to the Data Availability Statement at the start of the manuscript PDF file)?

Reviewer #1: Yes

Reviewer #2: Yes

Reviewer #3: Yes

4. Is the manuscript presented in an intelligible fashion and written in standard English?

Reviewer #1: Yes

Reviewer #2: Yes

Reviewer #3: Yes

5. Review Comments to the Author

Reviewer #1: Thank you for the submission. This is a research with a well-constructed method and execution. Its findings resonate to medical practice not only in low- and middle-income countries but all over the world where travel medicine is practiced. The limitations are also well-outlined.

Reviewer #2: This manuscript compares the diagnostic accuracy of four diagnostic methods for P. falciparum malaria and estimates P. falciparum prevalence in Kinshasa Province using each diagnostic method. The authors emphasize the importance of considering differences in assay performance when using surveillance data to guide policy decisions. The manuscript is well written and presents findings in a clear fashion.

Some minor comments to follow:

- Specify on Methods that samples used in this analysis were collected in March-June 2018, as indicated in the Abstract.

- I suggest to add how many individuals had available results for only some of the diagnostic assays. Were these individuals different than the 1431 individuals finally included?

- Please justify briefly why only HRP2 antigen positivity was utilized for the analysis, when more antigens were studied with BBA. Similarly, HRP2/pLDH RDTs were used, but only HRP2-ban positive results were reported (Table 1). Additional information could be interesting for the readers and added at least as supplementary material.

- Were samples run in duplicates/triplicates for the multiplex bead-based immunoassay?

- In page 12, line 228 there is a typo: “stratifiedhealth areas” instead of “stratified health areas”.

- In Table 1, I suggest to change “Age (years)” for “All age groups” in the row where overall numbers by health area are reported.

- In Table 1, authors refer to P. falciparum prevalence. However, any Plasmodium species visualized using microscopy was reported (as indicated in a footnote). Is information on species available? If yes, it could be used to restrict microscopy to those positive for P. falciparum. Are other species apart from P. falciparum important in Kinshasa Province?

- In Table 1 footnote, letter a is repeated and letter e is missing.

- In Figure 2B, consider deleting the names within the figure and do a legend (as in Figure 2A) for simplification.

- Diagnostic performance evaluation included sensitivity and specificity versus PCR/BBA/"alloyed gold standard", but other complementary measures could be used such as: positive predictive value, negative predictive value, positive likelihood ratio, negative likelihood ratio, diagnostic odds ratio, and area under the receiver operating characteristic curve. Graphically, Venn diagrams could be included to visualize number of infections detected by each method and the overlapping. I suggest adding some of these measures/graphs in the section “Diagnostic performance for the detection of P. falciparum”.

- In Table 3, could %s be added (apart from the Ns) in the lower and upper 95% confidence limit?

- In introduction or discussion, new ultrasensitive malaria RDTs could be mentioned and put into context.

Reviewer #3: The authors presented the malaria prevalence data of different diagnostic tests for malaria in varying malaria endemic area of Kinshasa province of Democratic Republic of Congo. Microscopy, bead-based-immunoassays, PCR and RDT were compared.

1. Introduction: Please add malaria case data of three different endemic areas of Kinshasa province from the latest government report.

2. Methods: Please provide number of household heads interviewed.

3. Line 124, it has been mentioned that “At the baseline visit of the second phase study…………………” but it has not been mentioned of endline in the manuscript.

4. Table 1, malaria prevalence based on age distribution; the age categories are four but there are five rows of data. If the first row is of the total, the sum of age distribution data is not equal to total in Bu and Kimpoko. Please correct data. Similarly, self-reported fever in last seven days, the total of the two categories is not 1431. I am wondering why the number of people above 25 years in the study were less as compared to other age groups.

5. In discussion, I suggest to add discussion on why there is variation in sensitivity and specificity of various diagnostic tests based on endemicity of malaria.

6. PLOS authors have the option to publish the peer review history of their article (what does this mean?). If published, this will include your full peer review and any attached files.

**Do you want your identity to be public for this peer review?** For information about this choice, including consent withdrawal, please see our Privacy Policy.

Reviewer #1: No

Reviewer #2: No

Reviewer #3: No

---

## [Decision Letter · Decision Letter 1]

15 Jun 2023

Impact of malaria diagnostic choice on monitoring of *Plasmodium falciparum* prevalence estimates in the Democratic Republic of the Congo and relevance to control programs in high-burden countries

PGPH-D-22-01821R1

Dear Alpha Oumar Diallo,

We are pleased to inform you that your manuscript 'Impact of malaria diagnostic choice on monitoring of *Plasmodium falciparum* prevalence estimates in the Democratic Republic of the Congo and relevance to control programs in high-burden countries' has been provisionally accepted for publication in PLOS Global Public Health.

Best regards,

Srinivasa Rao Mutheneni, PhD

Academic Editor

Reviewer Comments (if any, and for reference):

Reviewer's Responses to Questions

**Comments to the Author**

1. If the authors have adequately addressed your comments raised in a previous round of review and you feel that this manuscript is now acceptable for publication, you may indicate that here to bypass the “Comments to the Author” section, enter your conflict of interest statement in the “Confidential to Editor” section, and submit your "Accept" recommendation.

Reviewer #1: All comments have been addressed

Reviewer #2: All comments have been addressed

Reviewer #3: All comments have been addressed

2. Does this manuscript meet PLOS Global Public Health’s publication criteria? Is the manuscript technically sound, and do the data support the conclusions? The manuscript must describe methodologically and ethically rigorous research with conclusions that are appropriately drawn based on the data presented.

Reviewer #1: Yes

Reviewer #2: Yes

Reviewer #3: Yes

3. Has the statistical analysis been performed appropriately and rigorously?

Reviewer #1: Yes

Reviewer #2: Yes

Reviewer #3: Yes

4. Have the authors made all data underlying the findings in their manuscript fully available (please refer to the Data Availability Statement at the start of the manuscript PDF file)?

Reviewer #1: Yes

Reviewer #2: Yes

Reviewer #3: Yes

5. Is the manuscript presented in an intelligible fashion and written in standard English?

Reviewer #1: Yes

Reviewer #2: Yes

Reviewer #3: Yes

6. Review Comments to the Author

Reviewer #1: (No Response)

Reviewer #2: All comments and suggestions have been addressed.

Reviewer #3: None

7. PLOS authors have the option to publish the peer review history of their article (what does this mean?). If published, this will include your full peer review and any attached files.

**Do you want your identity to be public for this peer review?** For information about this choice, including consent withdrawal, please see our Privacy Policy.

Reviewer #1: **Yes: **Tinsae Alemayehu

Reviewer #2: No

Reviewer #3: **Yes: **Megha Raj Banjara
